# *Meyerozyma caribbica* Isolated from Vinasse-Irrigated Sugarcane Plantation Soil: A Promising Yeast for Ethanol and Xylitol Production in Biorefineries

**DOI:** 10.3390/jof9080789

**Published:** 2023-07-26

**Authors:** Bárbara Ribeiro Alves Alencar, Renan Anderson Alves de Freitas, Victor Emanuel Petrício Guimarães, Rayssa Karla Silva, Carolina Elsztein, Suzyanne Porfírio da Silva, Emmanuel Damilano Dutra, Marcos Antonio de Morais Junior, Rafael Barros de Souza

**Affiliations:** 1Laboratory of Biomass Energy, Department of Nuclear Energy, Federal University of Pernambuco, Recife 50670-901, Brazil; barbara.ribeiro.dbbs@gmail.com (B.R.A.A.); suzyaneeporfirio@gmail.com (S.P.d.S.); emmanuel.dutra@ufpe.br (E.D.D.); 2Laboratory of Microbial Genetics, Department of Genetics, Federal University of Pernambuco, Recife 50670-901, Brazil; renan_anderson@hotmail.com (R.A.A.d.F.); raayssa.karla@gmail.com (R.K.S.); carolinaelsztein@gmail.com (C.E.); 3Laboratory of Microbial Metabolism, Institute of Biological Sciences, University of Pernambuco, Recife 50110-000, Brazil; victor.petricio@ufpe.br

**Keywords:** cactus, hydrolysate, lignocellulose, mixed substrate, non-conventional yeast

## Abstract

The production of fuels and other industrial products from renewable sources has intensified the search for new substrates or for the expansion of the use of substrates already in use, as well as the search for microorganisms with different metabolic capacities. In the present work, we isolated and tested a yeast from the soil of sugarcane irrigated with vinasse, that is, with high mineral content and acidic pH. The strain of *Meyerozyma caribbica* URM 8365 was able to ferment glucose, but the use of xylose occurred when some oxygenation was provided. However, some fermentation of xylose to ethanol in oxygen limitation also occurs if glucose was present. This strain was able to produce ethanol from molasses substrate with 76% efficiency, showing its tolerance to possible inhibitors. High ethanol production efficiencies were also observed in acidic hydrolysates of each bagasse, sorghum, and cactus pear biomass. Mixtures of these substrates were tested and the best composition was found for the use of excess plant biomass in supplementation of primary substrates. It was also possible to verify the production of xylitol from xylose when the acetic acid concentration is reduced. Finally, the proposed metabolic model allowed calculating how much of the xylose carbon can be directed to the production of ethanol and/or xylitol in the presence of glucose. With this, it is possible to design an industrial plant that combines the production of ethanol and/or xylitol using combinations of primary substrates with hydrolysates of their biomass.

## 1. Introduction

The use of lignocellulosic biomass to produce biofuels, thermal and electrical energy, fertilizers, biomaterials, and high added value chemicals has been a strategy to reduce greenhouse gas emissions and reduce the negative impact of climate change [1]. It is estimated that 181.5 billion tons of lignocellulosic biomass are produced in the world, which clearly indicates the importance of this bioresource for humanity [2]. However, the technical, economic, social, and environmental feasibility of using lignocellulosic biomass has been questioned, especially when production arrangements operate for the production of few products with low added value. To improve these indicators, the concept of biorefineries emerged to convert all lignocellulosic biomass fractions and process by-products into energy, biofuels and high value-added bioproducts [3,4,5]. Solarte-Toro and Alzate [6] have recently published a very interesting review on the concept, design and critical points relating to biorefineries. Among the points of discussion raised, the use of multifeedstock appears as one of the relevant factors to be considered.

An example of biorefineries is the successful case of first-generation ethanol production in Brazil, where the predominant raw material is sugarcane with a harvest of approximately 631 million tons per year [7]. Its processing generates the production of sugar and ethanol (28 billion liters) and its by-products are bagasse (180 million tons), straw (203 million tons), filter cake (24 million tons) and vinasse (420 billion liters). These by-products are already used in the production of thermal and electrical energy, as fertilizer in the crop field and in the production of biogas [8]. Despite this, many studies seek to take advantage of the chemical composition of these biomasses to improve the production of biofuels and new products, such as studies with fractioning of sugarcane bagasse components [9,10].

In addition to sugarcane biomass, other sources of biomass are beginning to gain ground in some regions of Brazil, such as the production of ethanol from corn in the Midwest and research into sweet sorghum [11] in Northeast Brazil. All these biomasses have cellulose, hemicellulose, and lignin fractions in the composition of the plant cell wall and that can be used in biorefinery arrangements [2,3,6,12]. Recently, we have shown the potential of sweet sorghum bagasse as energy biomass that can be considered in this biorefinery context as well [13,14,15].

In addition to these conventional lignocellulosic biomasses, our inter-laboratory research group has also studied the potential of pectin-rich biomasses, such as those from the *Opuntia* and *Neopala* cacti, for ethanol production from the hydrolysis of the prickly pear cladodes [16,17]. The hydrolysis of its biomass generates a substrate rich in glucose that can be fermented by *Saccharomyces cerevisiae* at high yield [16]. However, this is no bagasse production in this case, which makes unprofitable the production of ethanol solely using this biomass. Therefore, the cactus hydrolysate serves as complement in mixture with more conventional substrates, such as sugarcane and sweet sorghum. The main advantages for the use of these substrates relies on the lower demand for water, especially the cactus, than sugarcane. It means that areas already degraded with low supply of water could be explored instead of natural areas and stop deforestations.

In addition to the search for different substrates, the concept of biorefineries also presupposes the search for strains of microorganisms capable of using the different sugars present in these substrates, as well as producing different products [2,3]. In the case of biomass such as bagasse and wood derivatives, the released xylose can be converted to ethanol or partially reduced to xylitol [2]. This use will depend on the microorganisms with capacity to metabolise this pentose. Several yeast species stand out as capable candidates for producing many of those products [18,19,20,21], such as the yeast *Meyerozyma caribbica* [22]. This is the new epithet of *Pichia caribbica* and the teleomorph of *Candida fermentati* (Division Ascomycota, class Saccharomycetes, order Saccharomycetales, family Debaryomycetaceae—Index Fungorum). This yeast has been isolated from soils around the world [23] and in the phylloplane of sugarcane leaves in different provinces of Thailand [24]. It was also isolated from rotting corn samples [22], among other niches worldwide. It was identified in the yeast population of the industrial fermentation processes for fuel-ethanol production in northeast Brazil that use sugarcane juice and/or molasses as substrate [25], as well as in the fermentation process to produce cachaça spirit from sugarcane juice substrate in stills in the state of Pernambuco, northern Brazil [26]. This is one the species fund in high abundance in coffee fermentation [27]. It was isolated from the nectar of tropical flowers of India, being the better xylitol producer of that osmotolerant yeast community [28], as well as from the ripe fruits of camu-camu plant from the Brazilian Amazon [29]. Therefore, this species presents a promissing biotechnological potential, serving as a biofactory to produce ethanol from cellulose-derived glucose and ethanol and/or xylitol from hemicellulose-derived xylose. The metabolic pathway taken by the cells, whether producing ethanol or xylitol, is directly linked to the availability of oxygen. The objective of this work was to evaluate the potential of the yeast *M. caribbica*, isolated in sugarcane soil irrigated with vinasse, to produce first- and second-generation ethanol as well as xylitol.

## 2. Materials and Methods

### 2.1. Soil Characteristics and Sampling

Soil samples were taken in March 2019 from sugarcane plantations in the surroundings of the Japungu Agroindustrial distillery, municipality of Santa Rita, Paraiba state, Brazil (latitude: −7.11631; longitude: −34.9812; south: 7° 6′ 59″; west: 34° 58′ 52″). The region is basically formed from an oxisols, generally composed by iron oxides, quartz and kaolinite, a clay mineral highly weathered, with average pluviometry of 115 (±6) mm and daily air temperature variation from 23 to 35 °C in the sampling period. The soil was constantly irrigated with the vinasse resulting from distillation of the fermented must. In that particular distillery of study, sugarcane harvesting goes from July to January and the fermentation process uses cane juice as substrate. However, the fermentation process extends to March mostly using the molasses coming from food sugar production as fermentation substrate. It means that at the time of sampling, the soil had been irrigated with molasses-based vinasse, whose average composition was: pH = 4.6 (±0.6), biochemical oxygen demand = 22.4 gO_2_/L (±3.6), chemical oxygen demand = 55 gO_2_/L (±14), total solids = 67.1 g/L (±20.3), volatile solids = 30 g/L (±14.1), total carbon = 17.5 g/L (±7.6), total nitrogen = 1.16 g/L (±0.64), C/N ratio = 16.3, potassium = 6.2 g/L (±2.28), sulfur = 5.06 g/L (±1.89), calcium = 4.87 g/L (±0.43), magnesium = 1.11 g/L (±0.58), phosphorus = 245 mg/L (±63.7) (data from the period of vinasse irrigation kindly provided by the distillery). It means that the yeasts in the soil were exposed and tolerant to high concentration of minerals, especially to potassium, sulfur and calcium in combination.

### 2.2. Soil Processing and Yeast Isolation

Samples were taken in ten different spots of each area, in 0–20 cm deep that were mixed in sterile plastic bags and transported to the lab in thermic boxes under refrigeration (2–8 °C). The amount of 50 g of the soil bland was suspended with 250 mL of Wallerstein Nutrient (WLN) broth (Merck KGaA, Darmstadt, Germany) containing a mix of the antibiotic ampicillin, chloramphenicol and nalidixic acid at 100 µg/mL each and cycloheximide at 5 mg/L in 500 mL flasks. The suspensions were incubated for 24 h at 30 °C under orbital agitation of 160 rpm for enrichment in non-*Saccharomyces cerevisiae* yeasts. Afterwards, the suspensions were submitted to serial dilutions with 0.85% sterile NaCl 0.9% *w*/*v* solution and 100 mL of each dilution were plated on synthetic medium containing 1.7 g/L yeast nitrogen base (YNB) without amino acids and ammonium (Sigma-Aldrich, St. Luis, MO, USA), 20 g/L glucose and 5 g/L ammonium sulphate, supplemented with antibiotics and cycloheximide as above. The plates were incubated for five days at 30 °C. The colonies were transferred for YPD plates (10 g/L yeast extract, 20 g/L peptone, 20 g/L glucose and 20 g/L bacto-agar) and incubated for more five days at 30 °C. Colonies were separated by morphological inspection and checked for purity.

### 2.3. Yeast Molecular Identification

Selected yeast colonies were identified by molecular analysis. Cells of each isolate were cultivated in YPD medium for 16–18 h at 30 °C at 160 rpm and used for DNA extraction [25]. Samples of one mL were transferred to 1.5 mL microtubes, and cells were harvested by centrifugation. The supernatant was throwed out and the cell pellets suspended with 600 µL of extraction buffer (200 mM Tris-HCl pH 8.0; 250 mM NaCl; 25 mM EDTA pH 8.0). Sodium dodecyl sulphate (SDS) was added to 10% (*w*/*v*) to each suspension and homogenised by vortexing. Then, the tubes were incubated at 65 °C for 10 min, with constant homogenisation. The lysates were centrifuged for 5 min at 10,000 rpm at 4 °C. A volume of 500 µL of the supernatants was transferred to new microtubes and mixed with 500 µL of extraction buffer and 10 µL Proteinase K solution (10 mg/mL). Following the incubation for 30 min at 56 °C, 425 µL of the lysate was transferred to the cartridges of DNA IQ Casework Pro Kit (Promega, Madison, WI, USA) for DNA purification in Maxwell^®^ 16 device (Promega) according to manufacturer instruction. Purified DNA was eluted with 50 µL elution buffer, quantified in Nanodrop equipment (Thermo scientific, Waltham, MA, USA) and checked for integrity in 1% agarose gel using 0.5x Tris-Borate-EDTA (TBE) buffer.

Yeast isolates were identified using molecular markers previously described [25,26]. Total DNA was subjected to amplification of the ITS1-5.8s-ITS2 locus using the primers ITS4 (5′-TCCTCCGCTTATTGATATGC-3′) and ITS5 (5′-GGAAGTAAAAGTCGTAACAA-3′) and the D1/D2 variable domains of the rDNA 26S gene using the primers NL1 (5′-GCATATCAATAAGCGGAGAAAAG-3′) e NL4 (5′-GGTCCGTGTTTCAAGACGG-3′) in 25 μL PCR reactions containing: 1.25 mM MgCl_2_, 10 mM each dNTP, 0.5 U Taq DNA polymerase, 5 mM each primer and 50 ng DNA. The cycling parameters were: initial denaturation at 94 °C for 5 min followed by 35 cycles of 94 °C/30 s for denaturation, 55 °C/1 min for annealing and 72 °C/1 min for polymerization, with final extension at 72 °C for 5 min. The amplicons were checked for integrity in 1% agarose gel electrophorese in 0.5× TBE buffer and then purified using Wizard^®^ PCR Preps DNA Purification System kit (Promega) following the manufacturer instructions. Samples were sent to the DNA sequencing and gene expression platform in the Centre of Biosciences of the Federal University of Pernambuco for sequencing in ABI Prisma 3500 devices. The chromatograms were analysed with the use of the BioEdit (v7.2.6) software and the clean nucleotide sequences used for yeast identification with the BLASTn tool in the GeneBank at the National Center for Biotechnology Information (NCBI) database (http://www.ncbi.nlm.nih.gov/BLAST (accessed on 10 July 2023)). Molecular identification used the criterion indicated for industrial yeast identification [26].

For phylogenetic analysis, these sequences were aligned in the online interface MAFT v.7 software [30] and manually edited in MEGA v.7 software. Afterwards, the sequences were submitted Bayesian analysis with that aid of MrBayes 3.2.7 software [31] available in the CIPRES Science Gateway portal [32], with the substitution models generated with MrModel 2.3 software [33]. Maximum-likelihood phylogeny was constructed using RAxML v7.2.8 6 available in XSEDE 8.2.12 accessed from the CIPRES Science Gateway portal [34]. Standard patterns were conducted with 1000 replicates of bootstrap. The phylogenetic tree was visualised in FigTree v.144 software.

### 2.4. Physiological Tests

Yeast isolates were cultivated in YPD as above and cell suspension were prepared in NaCl 0.9% *w*/*v* solution to around 1.0 absorbance unit at 600 nm (A600). Dilutions of 10× and 100× of the cell suspensions were prepared in NaCl 0.9% *w*/*v* solution and 10 μL of each dilution were spotted on selective plates composed of 1.7 g/L YNB plus a carbon source (20 g/L glucose or xylose) and a nitrogen source (5 g/L ammonium sulphate). To test for acid tolerance, yeast cells were inoculated in YNB containing glucose and ammonium sulphate and adjusted to different pH values with sulphuric acid (50% *v/v* solution).

The growth capacity of *M. carribica* was evaluated in different fermentation media composition (Table 1). For the assays, the yeast was previously grown in YPD medium (2% *w*/*v* glucose, 2% *w*/*v* peptone, and 1% *w*/*v* yeast extract) for 24 h. After that, the microbial suspension was centrifuged at 3600 rpm for 5 min. Afterward, the cells were washed three times with 0.9% *w*/*v* NaCl and centrifuged (3600 rpm; 5 min) to remove the dirty liquid between each wash. After that, the cells were transferred to the different substrates to the initial concentration of 0.1 A600. The cultivations were performed in orbital shakers for 24 h, at 30 °C and 200 rpm. Biomass production was quantified by direct inspection of media absorbance variation. All experiments were performed in biological triplicates.

### 2.5. Acid Hydrolysates Production and Molasses

Hydrolysates of sugarcane and sweet sorghum bagasses and Opuntia were produced by applying a solid load of 10 *w*/*v* for bagasse and 15% *w*/*v* for Opuntia. For this, a solution of 1.5% *v/v* of H_2_SO_4_ was used [35]. The flasks containing the biomass and the acid solution was kept in an autoclave for 30 min at 121 °C and 1 atm. After cooling, the suspension was centrifuged at 3600 rpm, for 5 min, with the liquid fraction, which is rich in carbohydrates, was collected and stored in a freezer at −20 °C. The molasses was kindly provided by the Ipojuca Agroindustrial distillery, located in the municipality of Ipojuca, Pernambuco, Brazil.

### 2.6. Fermentation Assays

Yeast cells were cultivated in YPB broth by successive batches to produce enough biomass. Then, they were collected by centrifugation and washed with NaCl 0.9% *w*/*v* solution. Fermentation substrates were composed by YNB broth containing 5 g/L ammonium sulphate and the carbon sources (40 g/L glucose or xylose alone or a mixture of glucose and xylose at 20 g/L each). Each substrate was inoculated with yeast biomass to around 100 g/L and fermentations were performed at 30 °C without agitation for up to 120 h. Samples were taken, centrifuged and the supernatant filtered in 0.22 μm Millipore filters. Every fermentation was performed in biological triplicates, with technical duplicates each. Metabolic flux distribution models were prepared according to Teles et al. [36], taking in account the input of carbon from the consumed sugar and the metabolic NADPH requirement to assimilate the consumed xylose.

Fermentation experiments using sugarcane juice, sugarcane molasses, plant biomass hydrolysates were performed as previously reported in our previous works [11,13,14,16,19,37,38]. In addition, mixed substrates were prepared according to Table 1 without nutritional supplementation. In brief, the yeast cells were cultivated in YPD medium at 30 °C and 200 rpm for 24 h. The cells were collected and resuspended in the same volume of fresh medium for further cultivation. This procedure was repeated until reaching enough biomass for the fermentation experiments. Then, the cells were collected and re-suspended in the fermentation substrate to initial biomass of 100 g/L. Static incubations at 30 °C lasted for different times according to the substrate. Whenever necessary, the flasks were slightly agitated at 70 rpm in rotatory shaker. Samples were withdrawn at indicated intervals, depending on the type of substrate, centrifuged and the supernatant stored for metabolites quantifications. *Saccharomyces cerevisiae* industrial strain JP1 was used as reference [11,13,16]. All experiments were performed in biological triplicates.

For the quantification of total nitrogen, 20 mL of the fermentative substrates were added in a test tube, which was attached to the apparatus. In a 125 mL Erlenmeyer flask, 10 mL of boric acid solution, with indicator, was pipetted. Into the distiller’s beaker, 10 mL of 13N NaOH will be pipetted. The sample was distilled to a volume of 50 mL. The distillate was titrated with 0.07143 N HCl until it turned from green to dark pink. Conductivity was determined by measuring with a conductivity meter. All experiments were performed in biological triplicates with technical triplicates each.

### 2.7. Substrate Detoxification

The detoxification of the sugarcane bagasse hydrolysates and substrate blends were performed with the application of biochar produced from the sugarcane bagasse. For this, the sugarcane bagasse was kept in a pyrolyzer for 1h at 600 °C. After that, the biochar produced was kept in hermetically sealed glass and protected from light. The detoxification assays were performed according to Ahuja et al. [39], with the following adaptations: 10% *w*/*v* biochar loading in the substrates and incubation shaker for 3 min at 100 rpm and 25 °C, followed by substrate centrifugation at 3600 rpm for 5 min. The detoxified substrates were recovered and used in the fermentation assays as above.

### 2.8. Effect of Acetate on Fermentation

Fermentation assays were carried out for evaluated the effect of acetic acid in the fermentative metabolism of *M. caribbica*. For that, experiments were performed using YP medium contend glucose and xylose as carbon sources in the proportion of 1:5 (total of 80 g/L of carbon sources). Yeast cell were pre cultivated in YPD medium at 30 °C and 200 rpm for 24 h. The cells were collected and resuspended in the YPDX medium contend 1.6 and 3 g/L of acetic acid to initial biomass of 50 g/L. The flasks were incubated at 30 °C with slightly agitation at 70 rpm in rotatory shaker. Samples were collected every 24 h for 120 h for metabolites quantifications. The assays were performed in biological triplicate.

### 2.9. Analytical Methods

Consumed sugar and other metabolites (glycerol, xylitol, acetic acid and ethanol) were identified by HPLC in Agilent Technologies 1200 Series device with a refractive index detector (RID), using HPX87H+ column (BioRad, Hercules, CA, USA) and mobile of 5 mM sulphuric acid at 45 °C. Calibration curves of standard metabolites were used for quantification and fermentative parameters were calculated.

### 2.10. Statistical Analysis

Values in the graphics and table represent the arithmetic mean (±standard deviation). The results were subjected to variance analysis (ANOVA) and were compared by the Tukey test (α = 0.05) using the software ASSISTAT v1.0 (Informer Technologies, Inc., https://assistat.software.informer.com/ (accessed on 10 July 2023)).

## 3. Results

### 3.1. Yeast Isolation and Identification

In the present work, we screened the yeast population in the soil of sugarcane plantations irrigated with vinasse, which turns those soils with high loads of minerals that can be toxic for most microorganisms. Following a series of dilutions and spreading in WLN medium, colonies of given morphotypes were isolated, checked for purity and submitted to molecular identification by 26S rDNA sequencing. Fives isolates were identified as *M. caribbica* (Figure 1). Phenotype quantitative tests showed that all *M. caribbica* isolates could grow aerobically on plates containing glucose or xylose as the single carbon source and the isolates W13, 53T2, RAC and RAF grew in medium adjusted to pH 2.5 with sulfuric acid. Then, the isolated RAC stood out as the most acid tolerant among the isolated yeasts and was chosen for further analysis. This isolated was deposited at the Department of Mycology Culture Collection (URM-Recife), Federal University of Pernambuco, which is part of the World Directory of Collections of Culture of Microorganisms (WFCC) under registration number 604 and can be released for research purposes upon request. This yeast was henceforth designated strain URM 8365 (Figure 1).

### 3.2. Ethanol Fermentation of Mineral Medium and Carbon Distribution Analysis

Fermentation assays in the mineral medium were carried out to evaluate the ethanol production by *M. caribbica* URM8365 from glucose and xylose as carbon sources. The results showed that all glucose was consumed and produced ethanol to a yield of 0.43 g/g and the glycerol was also detected as a by-product of fermentation (Table 2). On the other hand, all initial xylose was detected at the end of fermentation (Table 2). This result contrasts with the experiments of aerobic growth on a plate, revealing the incapacity of this yeast to assimilate xylose when the oxygen supply is limited or absent. Then, glucose and xylose were mixed to approximately the same initial concentration. In this case, glucose was exhausted while only 21.7% of the initial xylose was consumed (Table 2). No xylitol was detected under this oxygen-limitation condition, while ethanol was the unique fermentation product. However, the final ethanol yield was calculated as 0.55 g/g if only glucose uptake was taken into consideration, above the maximal theoretical of 0.51 g/g. It indicated that part of the consumed xylose was also transformed to ethanol.

To calculate how much of xylose was fermented, we simulated a metabolic model taking into consideration that the amount of 28 mmol (4.11 g) xylose that was consumed by the cells would require 28 mmol of NADPH for the first enzymatic step of xylose reductase that converts xylose to xylitol (Figure 2). This reducing equivalent can be produced by the deviation of part of glucose-6P to the Pentose Phosphate Pathway (PPP) or via the acetaldehyde dehydrogenase in the Pdh bypass pathway [36]. Since acetate was not detected (Table 2), then we considered that NAPDH was the majority produced by the PPP with the metabolization of 14 mmol of glucose (Figure 2). In this case, the sum of metabolites flowing directly through glycolysis with those metabolites returning from PPP would produce 181 mmol of ethanol, resulting in a yield of 0.49 g ethanol per gram of glucose. This is 13% above the yield calculated from glucose alone (Table 2). In parallel, the 28 mmol of xylose consumed would produce 14 mmol of ethanol from glyceraldehyde-3P. The theoretical production of 195 mmol (9.02 g) of ethanol and the ethanol yield of 0.43 g/g was in the range of the experimental ethanol concentration of 9.29 g/L (±1.73) and yield of 0.44 g/g (Table 2). Therefore, the ethanol yield from xylose was calculated as 0.16 g/g. The major concern about this metabolic adjustment regards the surplus of NADH produced by xylitol oxidation (Figure 2). Taking the fact that xylose has been consumed and no xylitol was detected, this unbalanced redox state was somehow surpassed by the yeast metabolism. Hence, it was plausible to suppose that the excess of 28 mmol of NADH might be used together with 14 mmol of fructose-6P from xylose and 38 mmol of ammonium from the medium to produce biomass. This hypothesis was experimentally supported by the production of twice more biomass at the end of fermentation in a sugar mix medium (2.09 g) than in a glucose medium (1.13 g) (Table 2). Thus, xylose consumption in oxygen limitation might be dependent on the presence of some glucose in the substrate.

### 3.3. Ethanol Fermentation of Industrial Substrates

Fermentation experiments were performed in molasses as the reference of industrial substrate for first-generation fuel ethanol (Figure 3A). Complete consumption of sucrose was achieved at six hours of fermentation, followed by glucose. Fructose concentration was raised during sucrose consumption, indicating that the cells secrete the enzyme invertase and, therefore, sucrose hydrolysis occurs in the medium (Figure 3A). All sugar was completely consumed at 72 h of fermentation to produce 46 g/L of ethanol and 5 g/L of glycerol (Figure 3A), with respective yields of 0.39 g/g and 0.04 g/g. It was compared to fermentation of a defined mineral medium used as reference of laboratory condition (Figure 3B). The results were physiologically like molasses after seven hours of fermentation: external hydrolysis of sucrose with transient fructose accumulation, ethanol yield of 0.38 g/g and glycerol yield of 0.04 g/g. These results indicated that *M. caribbica* URM 8365 can efficiently ferment molasses without suffering the interference of any possible inhibitors present at average levels in the substrate.

Afterwards, hemicellulose hydrolysates were tested as substrates for second-generation fuel ethanol. The hydrolysis produced substrates with xylose:glucose ratios of 7:1 for sugarcane bagasse (Figure 4A) and 2.5:1 for sweet sorghum (Figure 4B). In both cases, all glucose was consumed resulting in ethanol yields of 0.42 g/g and 0.45 g/g for sugarcane and sweet sorghum, respectively. These values were higher than those calculated for molasses and mineral medium. On the other hand, xylose remained untouched in these substrates. The explanation rested on the presence of the acetic acid in the hydrolysates, an already-known inhibitor of xylose metabolization. The concentration of this acid was between 2 to 3 g/L (34 to 50 mM) in the hydrolysates (Figure 4A,B). Thus, the challenge is to remove most of the acetate by physical procedures or to dilute it by mixing the hydrolysates with other substrates.

The glucose-rich cactus pear hydrolysate was fermented by *M. caribbica* URM 8365 cells. All glucose was consumed in six hours to produce 27 g/L of ethanol, with a final yield of 0.49 g/g.

### 3.4. Fermentation of Mixed Industrial Substrates

The results above showed that *M. caribbica* URM 8365 fermented biomass hydrolysates at even higher efficiency than molasses. However, it is not feasible to concept an ethanol industry based solely on these alternative substrates. Instead, the idea is to incorporate these hydrolysates into the conventional sugarcane-based matrix, especially considering the use of molasses produced from the crystal sugar milling process. Based on this, a series of combinations were prepared by mixing sugarcane molasses and bagasse hydrolysates (Table 1). The influence of these compositions on cell physiology was first tested in aerobic cultures (Figure 5). The results showed a higher cell growth in the molasses than in the hydrolysates, which might rely on the difference of nitrogen content since the hydrolysates were not supplemented with nutrients. In a way, it helped to explain the highest ethanol yields on the hydrolysates. The final yeast biomass achieved in the blends depends on the proportion of molasses in the mixture. The lowest growth was observed with the blends M4 and M5 (Figure 5), which might be the consequence of the high gravity condition above 300 g/L of sugar in the mixture. No significant difference in yeast growth was observed when the substrates were supplemented with ammonium, except for M4 and M5 where it at least doubled the final yeast biomass (Figure 5).

The fermentation profiles using these different blends showed that all hexoses were consumed by the yeast cells, except for glucose and fructose in M4 (Figure 6D) and M5 blends (Figure 6E). These sugars accumulated in the substrate probably due to the paralysis of the carbon flux by the central metabolism due to the imbalance of nitrogen, as indicated by the aerobic growth experiments (Figure 5). Ethanol was produced in all conditions almost proportional to the initial sugar concentration (Figure 6; Table 3). However, the calculated final ethanol production was lower when the sugar content was very high. The effect is particularly relevant when intending to ferment at high gravity with higher input of molasses. Xylose concentration was only relevant in M1 to M3 blend, where it surpassed 2.5 g/L, but it was not used by the yeast cells in fermentation (Figure 6A–C). The hypothesis in this case was the absence of oxygen in the fermentation condition. All xylose was consumed when the flasks were gently agitated at 70 rpm. in an orbital shaker resulting in xylitol production. Therefore, improvements were necessary to adequate the production of ethanol and xylitol in the concept of a biorefinery that uses any of these substrates.

To investigate the effect of the initial sugar concentration on the yeast fermentation capacity, we plotted the final ethanol concentration and final yield against the initial concentration of assimilable sugar taking from all fermentation conditions tested (Figure 7). The fifth-order regression curves indicated that a dissociation of ethanol production and ethanol yield up to 120 g/L of sugar. Taking in account the cost-benefit aspect of ethanol production, the best scenario for *M. caribbica* URM 8365 was the substrate containing 120 g/L of assimilable sugar even an ethanol yield of 0.4 g/g, meaning 78% of batch fermentation efficiency (Figure 7).

### 3.5. Xylitol Production

In addition to ethanol, the assimilation of xylose can also produce xylitol. Similar to what was observed in mineral medium (Table 2), fermentation of xylose-rich hydrolysates did not produce xylitol when the cultures were left static, with severe limitation, or even absence of oxygen. Then, we simulated a condition in which the mineral medium presented xylose:glucose ration similar to found in the bagasse hydrolysates. At this condition, xylitol was only produced when the cultures where gently agitated at 70 rpm (Table 3). However, low xylitol production was observed with a yield of 0.18 g/g. It is noteworthy the fact that ethanol production is much higher than expected for the amount of glucose consumed (Table 3). This indicated that part of the xylose was also converted to ethanol in this sugar mixture, as seen in Table 2. From these values, it was possible to determine the distribution of xylose carbon for the two products. Considering that each xylitol molecule comes from a reduced xylose molecule, then the production of 11.7 g of xylitol used 11.7 g of the total 63.6 g of xylose consumed, leaving 51.9 g of xylose. Considering the calculated yield of 0.43 g/g for the conversion of glucose to ethanol (Table 1; Figure 2), 4.77 g of ethanol were produced from glucose and the remaining 6.9 g of ethanol came from xylose. In the conservative perspective of the calculated yield of 0.18 g/g of the conversion of xylose to ethanol (Table 2), then 38.3 g of the remaining 51.9 g of xylose were converted to ethanol. Finally, the remaining 13.6 g of xylose must have been used for biomass generation. This result shows that the presence of glucose actually stimulates the conversion of xylose to ethanol, impairing the production of xylitol.

Afterwards, fermentation assay in synthetic medium containing xylose and glucose were performed in the presence of the acetic acid (1.6 g/L and 3.7 g/L), according to the concentration found in bagasse hydrolysate. Interestingly, the condition with 1.6 g/L of acetic acid decreased only the parameters related to xylose metabolism, while glucose consumption and total ethanol production were not affected (Table 3). Using the same rationale above, it was possible to calculate that 7.41 g of ethanol came from consumed xylose. This value is even higher than that calculated for the medium without acetic acid. However, xylitol production was reduced by 58%. Since the conversion of xylose to ethanol metabolically involves its initial reduction to xylitol, it makes no sense to consider that acetic acid reduced the ability of the yeast to produce xylitol. Thus, these data showed that this acid at this mild concentration accelerated the complete metabolization of xylose before xylitol left the cell.

In addition, the highest acetic acid concentration completely inhibited the xylose metabolism, with slight glucose consume (Table 3). Indeed, the presence of acetic acid in the hydrolysates may be also of major concern, since its presence in concentrations above 1.5 g/L impaired xylose consumption from synthetic medium, sugarcane, and sorghum bagasse hydrolysates. Thus, fermentations were performed with sugarcane hydrolysates (5 xylose: 1 glucose) that were detoxified with the aid of active coal. This treatment reduced the content of acetic acid in the hydrolysates from 2.5 g/L to 0.6 g/L. Ethanol yield from glucose remained around 0.44 g/g and was not affected by the detoxifying treatment. In addition, xylitol was detected at 6 g/L in the detoxified substrate (Y = 0.3 g/g), which hugely contrasted with its absence in fermentation of untreated hydrolysates.

## 4. Discussion

In the present work, we studied a strain of *M. caribbica* isolated from a vinasse-irrigated soil used for sugarcane cultivation in Northeast Brazil. This yeast has been isolated from soils, cereals, nectar and industrial processes around the world. The isolation of this yeast from these spots reveals the adaptation of this species in the environment of sugarcane plantation and processing. The phylogenetic analysis performed based on 26S rDNA sequencing discriminated *M. caribbica* within the heterogeneous *Meyerozyma guilliermondii* complex (Figure 1), in special from its closed-related species *M. guilliermondii* [40,41].

The selected strain *M. caribbica* URM 8365 used glucose or xylose for aerobic growth, but only glucose for ethanol fermentation when oxygen was very limited or absent (Table 2). Neither xylose was consumed nor xylitol was produced by *M. caribbica* under conditions of severe oxygen limitation. On the other hand, Tadioto et al. [22] reported that up to 32% of xylose was converted to xylitol when cultures were sufficiently aerated. Indeed, xylose assimilation seems to be not an easy task for yeast cells. *Pichia stipitis* cells consumed around 66% of xylose after 72 h of fermentation [42] and *M. caribbica* consumed all xylose only after 120 h of cultivation [43], despite the experiments used in rich medium. On the other hand, Veras et al. [44] reported high ethanol yields with full xylose consumption by *Scheffersomyces stipitis* and *Spathaspora passalidarum* in mineral medium under oxygen-limited conditions.

The results from mixed media in the present work showed that xylose is consumed during fermentation and part of this sugar is converted to ethanol when glucose is present in the medium. That is, the impediment of xylose fermentation seems to be due to the cellular unavailability of cofactors reduced in the absence of oxygen. Lane et al. [45] improved the co-assimilation of glucose and xylose by slowing down the activity of hexokinase and glucokinase enzymes. Later, Trichez et al. [46] showed that evolved strains of *S. stipitis* and *S. passalidarum* harbouring a mutant allele of the glucose transporter *HXT2.4* gene had improved co-assimilation of glucose and xylose. From the energetic point of view, the glucose consumed in the present work corresponded to 5.6 mM, which could theoretically produce up to 11.2 mM of NADPH from the PPP. Stoichiometrically, one mol of xylose requires one mol of NADPH to be reduced to one mol of xylitol. Since xylitol was detected at 76.3 mM final concentration, it indicated that the required NADPH originated not only from the PPP, but also from some other metabolic oxidative reactions capable of providing the extra 65.1 mM of reducing power. This high requirement of reduced cofactor indicated that NADH, both glycolytic and from TCA, might also be requested for the xylose metabolism.

Besides the energetic requirement, there are other concerns regarding xylose assimilation in the central carbon metabolism, such as the reported inhibition by acetic acid [47]. Indeed, acetic acid was the major inhibitor of xylose fermentation for *M. caribbica* when using hemicellulose hydrolysates (Figure 4). This organic acid can induce intracellular acidification that increases the demand for ATP and impairs the metabolic flux through the central metabolism [48]. The xylitol production by *P. stipitis* NCIM 3497 was also inhibited in the presence of 3.7 g/L of acetic acid [42]. Tadioto et al. [22] also reported the impairment of xylose consumption and xylitol production by *M. caribbica* from stalk-straw corn hydrolysates that contained acetic acid above 3 g/L. In the present work, xylitol production was impaired at lower concentration of 2.5 g/L of acetic acid in the bagasse hydrolysates. In addition, we observed the production of xylitol to a yield of 0.3 g/g in not-supplemented hydrolysates after 75% removal of acetic acid with active coal. Nagarajan et al. [43] also detoxified the hydrolysate with activated coal reducing from 9 g/L to 3 g/L the content of acetic acid in the substrate and increasing xylitol yield from 0.49 g/g to 0.54 g/g together with the increment of the yeast biomass formation. However, like many other reports in the literature, the authors supplemented the detoxified hydrolysate with salts and yeast extract, reaching higher yields and titers. Substrate supplementation always impose higher production costs, which are for many times impeditive for the production of commodities like ethanol and low- to middle-add value products like xylitol. Here, although showing lower yield, we used conditions that would be closer to the industrial process. Therefore, in the case of the biorefinery concept, part of the bagasse could be treated more mildly to produce a hydrolysate with the appropriate proportion of xylose and glucose for the optimal xylitol production, without the presence of acetic acid in the inhibitory concentration.

In conclusion, the results obtained in the present work showed the potential of *M. caribbica* URM 8365 strain for 1stG and 2ndG ethanol and xylitol production using industrial substrates: ethanol achieving more than 75% of fermentation efficiency for molasses, sugarcane juice and sugarcane and sorghum bagasse hydrolysates. The highest concentration of ethanol of 50 g/L was obtained in the M3 medium, which is composed of 25% molasses and 75% of biomass hydrolysate. In this condition, devoid of nutritional supplementation, there seems to present the best proportion and concentration of fermentable carbohydrates. Moreover, unlike the fermentation tanks that will receive a stream of juice, molasses or cellulose hydrolysate, the tanks for stream of hemicellulose hydrolysate must ensure a minimum of aeration for conversion of xylose into ethanol and xylitol. More studies are necessary to optimize the production process in order to integrate ethanol and xylitol in the same industrial plant.

## Figures and Tables

**Figure 1 jof-09-00789-f001:**
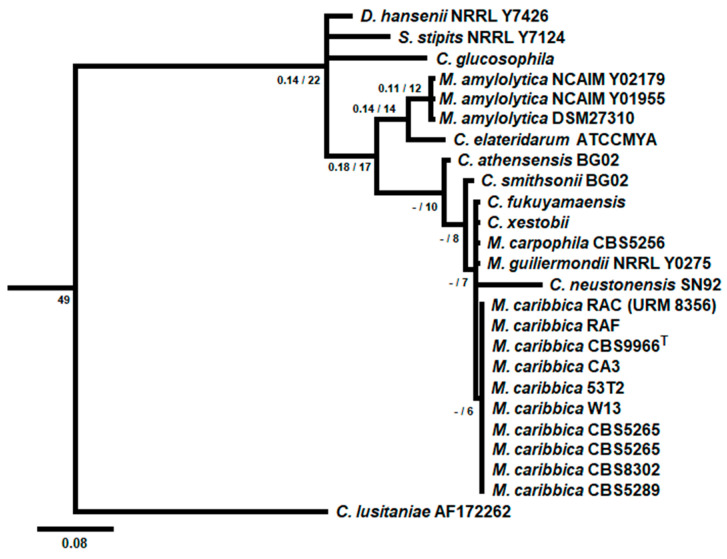
Phylogenetic analysis based on the nucleotide sequence of D1/D2 domain of the 26S rRNA gene for the identification of yeast isolated from the vinasse-irrigated sugarcane plantation soils.

**Figure 2 jof-09-00789-f002:**
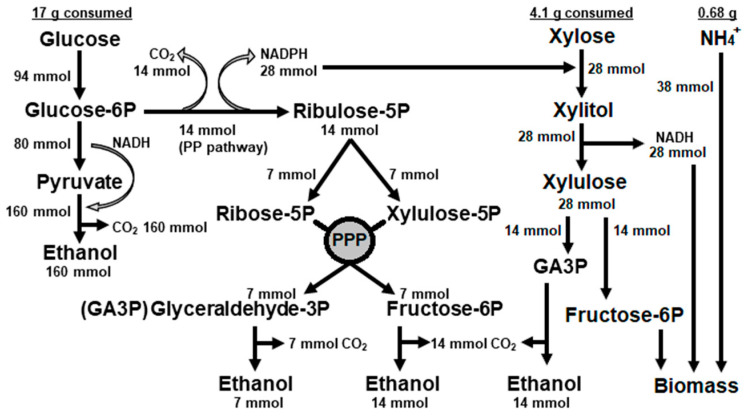
Metabolic model for carbon distribution in the central metabolism of *Meyerozyma caribbica* URM 8365 based on the physiological parameters calculated from the fermentation assays in mineral medium containing mixture of glucose and xylose. Stoichiometric calculations were performed according to Teles et al. [33].

**Figure 3 jof-09-00789-f003:**
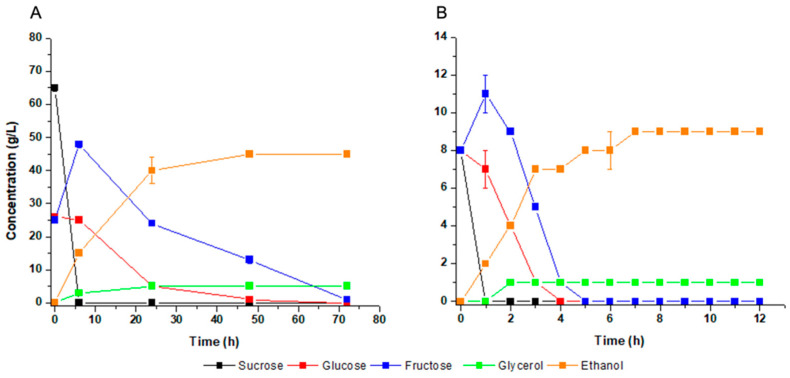
Fermentation profile of *Meyerozyma caribbica* URM 8365 in sugarcane molasses (**A**) and in mineral medium containing sucrose (**B**).

**Figure 4 jof-09-00789-f004:**
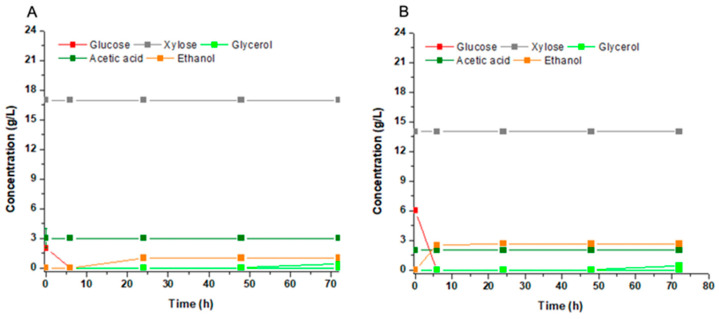
Fermentation profile of *Meyerozyma caribbica* URM 8365 in acid hydrolysates of sugarcane (**A**) and sweet sorghum (**B**) bagasses.

**Figure 5 jof-09-00789-f005:**
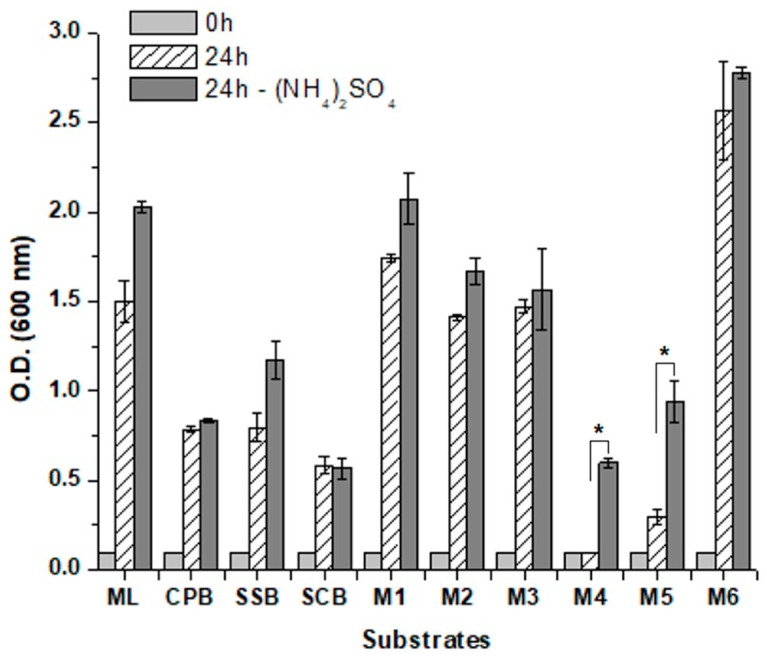
Aerobic growth of *Meyerozyma caribbica* URM 8365 in sugarcane molasses (ML), in hydrolysates of cactus pears (CPB), sugarcane (SCB) or sweet sorghum (SSB) and different mixtures of these substrates (M1 to M6) as described in Table 1. Optical density of the cultures was measured at the beginning (light grey columns) and after 24 h (striped columns) and 48 h (dark grey columns). * Significant difference at α < 0.01.

**Figure 6 jof-09-00789-f006:**
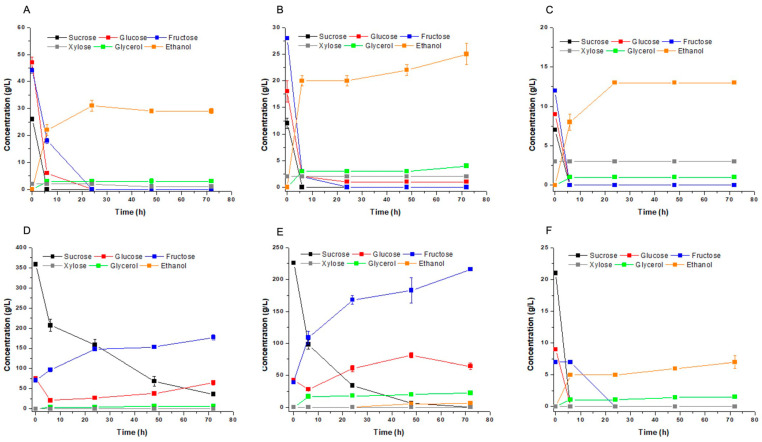
Fermentation profile of *Meyerozyma caribbica* URM 8365 in different mixtures of substrates M1 (**A**), M2 (**B**), M3 (**C**), M4 (**D**), M5 (**E**) and M6 (**F**) prepared as described in Table 1.

**Figure 7 jof-09-00789-f007:**
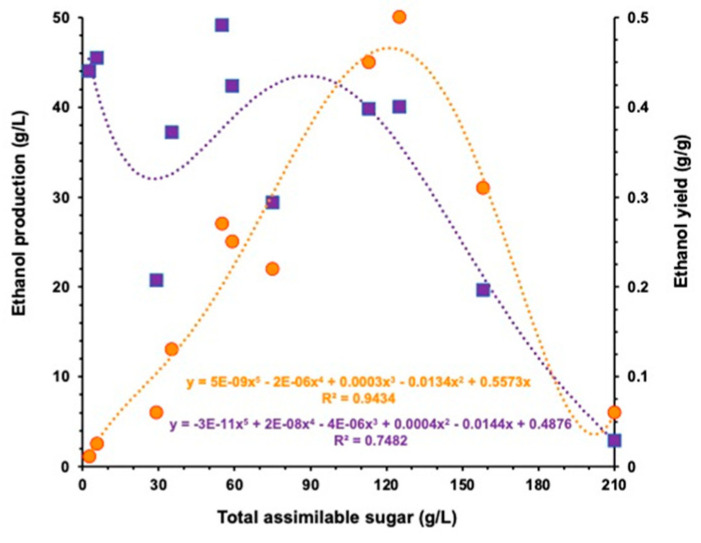
Correlation between the initial sugar concentration and the ethanol production or the ethanol yield by *Meyerozyma caribbica* URM 8365 as calculated from the fifth-order correlation curves.

**Table 1 jof-09-00789-t001:** Composition of fermentation substrates and blends.

	TFC	Total N		Conductivity	Composition			
Substrate	(g/L)	(%)	C/N	(ms/cm)	ML	SCB	SSB	CPB
ML	107.51	4.01	160.8	33.1	100%	--	--	--
SCB	21.27	1.81	70.5	122.0	--	100%	--	--
SSB	17.96	1.96	55.0	109.8	--	--	100%	--
CPB	64.09	4.21	91.4	57.0	--	--	--	100%
M1	127.42	3.21	238.2	54.7	70% *	10%	10%	10%
M2	94.66	3.31	171.6	64.8	50% *	16.7%	16.7%	16.6%
M3	52.31	2.46	127.5	83.8	25% *	25%	25%	25%
M4	306.97	6.76	272.5	29.9	50%	16.7%	16.7%	16.6%
M5	108.8	5.11	127.7	56.4	25%	25%	25%	25%
M6	36.63	3.26	67.4	93.4	10%	30%	30%	30%

Abbreviations: TFC—Total fermentable carbohydrates; ML—Molasses; SCB—sugarcane bagasse hydrolysate; SSB—sweet sorghum bagasse hydrolysate; CPB—cactus pear biomass hydrolysate. * Molasses was diluted six times before being mixed with the SCB, SSB, and CPB hydrolysates.

**Table 2 jof-09-00789-t002:** Fermentative parameters of *Meyerozyma caribbica* URM8365 isolated from vinasse-irrigated sugarcane plantation soil in synthetic medium containing different sugars.

Parameter	Glucose	Xylose	Glucose + Xylose
Initial glucose (g L^−1^)	37.85 (±0.90)	-	17.00 (±0.23)
Initial xylose (g L^−1^)	-	38.82 (±0.96)	18.95 (±0.20)
Final glucose (g L^−1^)	0.0 (±0.00)	-	0.0 (±0.00)
Final xylose (g L^−1^)	-	38.62 (±0.99)	14.84 (±0.31)
Ethanol produced (g L^−1^)	16.19 (±1.09)	0.00	9.29 (±1.73)
Glycerol produced (g L^−1^)	0.67 (±0.12)	0.00	0.00
Ethanol yield (g/g)	0.43 (±0.03)	0.00	0.44 (±0.04)
Glycerol yield (g/g)	0.02 (>0.00)	0.00	0.00
CO_2_ yield (g/g) *	0.41	0.00	0.42
Biomass yield (g/g)	0.03	0.00	0.09
Carbon balance (%)	89%	0.00	95%

* Stoichiometrically calculated from ethanol [36].

**Table 3 jof-09-00789-t003:** Fermentation parameters of *Meyerozyma caribbica* URM 8365 in mineral medium containing a mixture of glucose and xylose in the absence or presence of acetic acid.

Parameters	Mineral Medium	+Acetic Acid 1.6 g/L	+Acetic Acid 3.7 g/L
Xylose consumed (g/L)	63.58 ± 5.26	50.30 ± 3.05	0.00
Glucose consumed (g/L)	11.10 ± 0.69	11.74 ± 0.966	2.50 ± 0.63
Ethanol produced (g/L)	11.67 ± 1.01	12.41 ± 1.33	0.00
Xylitol produced (g/L)	11.67 ± 1.1	4.85 ± 0.57	0.00
Xylitol yield (g/g)	0.18 ± 0.04	0.1 ± 0.01	0.00

## Data Availability

Not applicable. All data is contained within the article.

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
