# Peer review of "Meyerozyma caribbica Isolated from Vinasse-Irrigated Sugarcane Plantation Soil: A Promising Yeast for Ethanol and Xylitol Production in Biorefineries"

_jof, 2023, doi:10.3390/jof9080789_

Round 1

Reviewer 1 Report

This manuscript  focuses a strain of Meyerozyma caribbica om the soil of sugarcane irrigated with vinasse.  This collection site is very special, and the yeast could produce ethanol and l xylitol. However, both the mechanism and the ability of ethanol and l xylitol production lacks sufficient innovation.

Major critical points:

1 The ethanol production level of the strain from Gluose is very low , the authors should improve the fermentation process.

2 Form Table 2 why this strain can not produce ethanol from xylose, the authors should provide more solid evidences.

3 The ability of co-utilizing glucose and xylose should be studied in detail. 

Author Response

This manuscript  focuses a strain of Meyerozyma caribbica om the soil of sugarcane irrigated with vinasse.  This collection site is very special, and the yeast could produce ethanol and l xylitol. However, both the mechanism and the ability of ethanol and l xylitol production lacks sufficient innovation.

Response: sorry, we did not understand the statement. Is the reviewer saying that the study of ethanol and xylitol is not necessary anymore? Or by using the words m'mechanism" and "ability", is the reviewer point that nothing is new on the study of the biochemistry of ethanol and xylitol production? 

Major critical points:

1 The ethanol production level of the strain from Gluose is very low , the authors should improve the fermentation process.

Response: In Table 2 we showed that our M. caribicca strain produce ethanol to yields of 0.43-0.44 g/g in mineral media, 0.42-0.45 g/g in bagasse hydrolysates and 0.49 g/g in cactus hydrolysate, which are in the range of S. cerevisiae. We do not understand how it can be considered low. Maybe this analysis was focused in the fermentation of molasses and molasses-derivative blands, which indeed were lower than we expected. In this case, the challenge is to adapt the cells to that mineral-rich substrate. It affects many non-S. cerevisiae strains, not only ours. 

2 Form Table 2 why this strain can not produce ethanol from xylose, the authors should provide more solid evidences.

Response: sorry, the evidence was provided with the results in mixed substrates in the same table. That experiment was performed under severe limitation or even absence of oxygen. Then, the cells lacked the supply of NADH necessary for the first enzyme reaction of xylose dehydrogenase. This was provided when glucose was present in the mixed medium. 

3 The ability of co-utilizing glucose and xylose should be studied in detail. 

Response: in the figure 2 we provided the calculation of carbon distribution and reduced cofactors supply that explain the connection between glucose and xylose metabolism. Then, in figure 4 we showed that acetate jeopardises this connection and impairs xylose-glucose co-consumption. Further, we showed that xylose consumption in the presence of glucose is increased by medium agitation, meaning that oxygen is necessary mainly for xylitol production. And, at last, we showed in Table 3 the influence of acetic acid in ethanol and xylitol production from glucose-xylose mixed substrate.

Additional response to the review report form:

a) The reviewer pointed that the research design "must be improved". In this case, it is imperative to state what is wrong with the experimental aproaches, what experiments are missing, what are unnecessary and what are uncorrectly performed. It may help us to do the neccessary adjustments.

b) the reviewer pointed that the connection beteween results and conclusions "mut be improved". In this case, it is also imperative to be clear about this lack of logical linkage. What conclusion did not find base on the experimental facts (results)?

c) the same analysis is applicable for the items introduction background, methodology adequacy and result presentation. Since the reviewer marked as "can be improved", we guess that it is not compulsory. However, we will be very grateful if the reviewer might be more specific in the correction for us to proceed with the necessary corrections.

Reviewer 2 Report

The manuscript titled Meyerozyma caribbica isolated from vinasse-irrigated sugar-2 cane plantation soil: a promising yeast for ethanol and xylitol 3 production in biorefineries” is focused on interesting topic: the production of fuels from renewable sources in particular new substrates and the search for microorganisms with different metabolic capacities. Was investigated the use of M. carribbica isolated in sugarcane soil irrigated with vinasse, to produce first- and second-generation ethanol as well as xylitol. The topic is important for technical, economic, social, and environmental aspects.

however, there are some aspects to improve the manuscript regarding the form and the structure. In particular, I suggest showing the data of all yeast’s species isolated in the study to better explain the choice of this specie. Moreover, this would give more value to the work done. I think that not showing the data of the other yeasts would deprive the manuscript of an important part.

Introduction

Line 72: change S. cerevisiae with Saccharomyces cerevisiae the first time it appears in the text it must be written in full

Line 85: change Meyerozyma caribbica in italics

The introduction in well written but I suggest adding some information regarding to M. carribica

Materials and Methods

Line 116-128: why did you use Waller-116 stein Nutrient (WLN) broth? I think that you could use a YPD broth. Moreover, why did not use a WL nutrient agar for the plate count? This medium was able to identify a different yeasts species. I think that YPD medium is not it is not suitable for a morphological identification, with this method I believe you may have lost some yeast species. How many colonies tested? It is important that you reported the number of isolates to give a significantly of the experiment.

Line 120: change non-Saccharomyces cerevisiae yeasts in italics

Line 194: change H2SO4 in H2SO4

Line 223: change Saccharomyces cerevisiae in S. cerevisiae italics. Please check and change in all manuscript

Line 260: change 45oC in 45°C

Results

Section 3.1: in M and M. you reported that you evaluated 10 different spots of each area, therefore I suggest reporting a table with a yeast species identified for each sample. In this way, we have an information regarding the yeast’s composition.

Check the italics form in all manuscript

Why did you not report all yeasts tested? it could be interesting to show the data relating to the strains tested and the difference between them.

Check the resolution of the figure, there are many mistakes regarding the style of the species and genera. I suggest reading the manuscript carefully and correct all form errors. Moreover, I suggest that the authors include data on the other species tested to improve understanding of the manuscript.

Author Response

The manuscript titled “Meyerozyma caribbica isolated from vinasse-irrigated sugar-2 cane plantation soil: a promising yeast for ethanol and xylitol 3 production in biorefineries” is focused on interesting topic: the production of fuels from renewable sources in particular new substrates and the search for microorganisms with different metabolic capacities. Was investigated the use of M. carribbica isolated in sugarcane soil irrigated with vinasse, to produce first- and second-generation ethanol as well as xylitol. The topic is important for technical, economic, social, and environmental aspects.

however, there are some aspects to improve the manuscript regarding the form and the structure. In particular, I suggest showing the data of all yeast’s species isolated in the study to better explain the choice of this specie. Moreover, this would give more value to the work done. I think that not showing the data of the other yeasts would deprive the manuscript of an important part.

RESPONSE: we think there must be a misunderstanding. We did not report different species, but different specimens of the same species: five isolates beloging to M. caribbica species. 

Introduction

Line 72: change S. cerevisiae with Saccharomyces cerevisiae the first time it appears in the text it must be written in full

RESPONSE: done

Line 85: change Meyerozyma caribbica in italics

RESPONSE: done

The introduction in well written but I suggest adding some information regarding to M. carribica

RESPONSE: we added more three references and made changes in the order of citation to add more information about this yeast

Materials and Methods

Line 116-128: why did you use Waller-116 stein Nutrient (WLN) broth? I think that you could use a YPD broth. Moreover, why did not use a WL nutrient agar for the plate count? This medium was able to identify a different yeasts species. I think that YPD medium is not it is not suitable for a morphological identification, with this method I believe you may have lost some yeast species. How many colonies tested? It is important that you reported the number of isolates to give a significantly of the experiment.

RESPONSE: we used with success that medium to identify the yeast species (almost 30 species) from ethanol fermentation (refs. 26 and 27). Thus, we repeated with samples of soil. No medium is absolutely efficient to discriminate yeasts based on colony morphology. If different species produce different type of colonies, most of them can be for sure discriminated on YPD plates. We have such experience from our results to identify yeasts from sugarcane fermentation for biofuel and cachaça. Nevertheless, itt proved enough for this work. We do not see the purpose of testing the significance of the experiments. This is a paper dedicated to explore the potencial of one of the yeast isolated from vinasse-irrigated soils. The other species we identified, such as Rhodotorula and Candida species, did not assimilate xylose. Thus, they were not important for this work. 

Line 120: change non-Saccharomyces cerevisiae yeasts in italics

RESPONSE: done

Line 194: change H2SO4 in H2SO4

RESPONSE: done

Line 223: change Saccharomyces cerevisiae in S. cerevisiae italics. Please check and change in all manuscript

RESPONSE: done

Line 260: change 45oC in 45°C

RESPONSE: done

Results

Section 3.1: in M and M. you reported that you evaluated 10 different spots of each area, therefore I suggest reporting a table with a yeast species identified for each sample. In this way, we have an information regarding the yeast’s composition.

RESPONSE: in the first sentence of section 2.2 we said that all tten samples were mixed. Thus, there is no information on the specific yeast composition.

Check the italics form in all manuscript

RESPONSE: done

Why did you not report all yeasts tested? it could be interesting to show the data relating to the strains tested and the difference between them.

RESPONSE: we chose the most acid tolerant  strain, since we intended to test the fermentation of acid-treated bagasses. There was no purpose in testing other strains that could not survive in this type substrate. 

Check the resolution of the figure, there are many mistakes regarding the style of the species and genera. I suggest reading the manuscript carefully and correct all form errors. Moreover, I suggest that the authors include data on the other species tested to improve understanding of the manuscript.

RESPONSE: done

Additional responses to the review report form:

a) The reviewer pointed that the introduction and references "must be improved". In this case, it is imperative to state how it can be done. What information are missing, what are unnecessary and what are uncorrectly performed. It may help us to do the neccessary adjustments. New references were added.

b) the reviewer pointed that experimental design, method description and results "can be improved". Thus, we guess that it is not compulsory, but we will be pleased to do the corrections as soon as the reviewer points how and what can be improved.

Round 2

Reviewer 1 Report

This manuscript  focuses a strain of Meyerozyma caribbica on the soil of sugarcane irrigated with vinasse.  This collection site is very special, and the yeast could produce ethanol and xylitol. However, both the mechanism and the ability of ethanol and l xylitol production lacks sufficient innovation.  I  means that there is no new mechanism and the titers of  ethanol and xylitol proudced by this strain did not show  significant advantages. Therefore, the authors should revised this manuscript and provide notable highlights.

Major critical points:

1 The ethanol production level of the strain from Gluose is very low , the authors should improve the fermentation process. In this part, I mean the concentration of ethanol (g/L ) is very low, not the yeild (g/g). The data of this study did not show the application potential for ethanol production.

2 The ability of co-utilizing glucose and xylose should be studied in detail.  The fermentation of glucose and xylose is important research content in this study. But the basic ability of this strain for co-utilizing glucose and xylose has not been investigated. The authors should investigete the ability growth and ethanol producion from glucose and xylose with different ratios and disscuss the  mechanisms.

3 The potential and costs of this strain for industrial application should be dissussed.

4  This manuscript point out " a promising yeast for ethanol and xylitol production". But what the logical linkage between ethanol and xylitol? Is the strain possess outstanding ability for xylose utilization? This manuscript did not show some common features of the strain benified for ethanol and xylitol production.

Author Response

1 The ethanol production level of the strain from Gluose is very low , the authors should improve the fermentation process. In this part, I mean the concentration of ethanol (g/L ) is very low, not the yeild (g/g). The data of this study did not show the application potential for ethanol production.

RESPONSE: The ethanol production by the yeast depends on the concentration of sugar available and consumed from the substrate. For example, in Table 2 we showed that 16 g of ethanol were produced from 38 g of glucose in synthetic medium. This yield of 0.43 would result in 52 g of ethanol from 120 g of sugar, which is in the range of S. cerevisiae strains. In addition, we tested different substrates mimicking the industrial process, which is harder than using lab media. In Figure 3 we reported the production of 45 g of ethanol from 115 g of sugars in molasses, without any supplementation. This yield of 0.39 would result in 55 g of ethanol if we increase sugar content to 140 g as they use in distilleries. Again, we are in the range of S. cerevisiae regarding hexoses. When we mixed the substrates bagasse hydrolysate and molasses, which represents a even harder condition for the cells, we produced ethanol at different levels. So, we plotting all the results, we concluded that “The fifth-order regression curves indicated that a dissociation of ethanol production and ethanol yield up to 120 g/L of sugar. Taking in account the cost-benefit aspect of ethanol production, the best scenario for M. caribbica URM 8365 was the substrate containing 120 g/L of assimilable sugar even an ethanol yield of 0.4 g/g, meaning 78% of batch fermentation efficiency (Figure 7).” In this case, we worked on the optimization of sugar concentration to improve ethanol fermentation for this yeast. In the work of Kruasuwan et al (2023) (DOI: 10.1007/s13205-022-03436-4) they reported the use a evolved strain of S. cerevisiae that fermented molasses at 0.40 to produce 65 g of ethanol from 162 g of consumed sugar. There are other examples like this in the literature attesting the difficulty of S. cerevisiae in fermenting molasses. Therefore, we do not consider that the level of fermentation by M. caribbica URM 8365 is low. In addition, M. caribbica was submitted to lignocellulosic hydrolysates that were not supplemented with yeast extract and/or peptone. Those substrates have a lot of inhibiting compounds that are harmful to the yeast. So, certainly there are species and strains of yeast that can be more efficient to cope with stresses and produce more ethanol, such as Saccharomyces cerevisiae, but the purpose of this edition is to explore and describe new yeast species with potential to produce biofuels and bioproducts. In our previous paper by Souza et al (2018) [Production of ethanol fuel from enzyme-treated sugarcane bagasse hydrolysate using D-xylose fermenting wild yeast isolated from Brazilian biomes. 3 Biotech 8: 312], we compared in the table 5 the results S. passalidarum fermenting bagasse hydrolysates. In our experiments, the yeast produced 24 g of ethanol from 70 g of consumed sugar, without any supplementation. In other papers, the authors supplemented the substrates with peptone and/or yeast extract to produce a bit more of ethanol. In the present work, M caribbica showed such low fermentation performance in bagasse hydrolysate without supplementation and in the presence of inhibitor. Once again, we considered that our yeast in the same level of other yeast reported in the literature for the fermentation of this type of substrate.

2 The ability of co-utilizing glucose and xylose should be studied in detail. The fermentation of glucose and xylose is important research content in this study. But the basic ability of this strain for co-utilizing glucose and xylose has not been investigated. The authors should investigete the ability growth and ethanol producion from glucose and xylose with different ratios and disscuss the mechanisms.

RESPONSE: We appreciate the review and points highlighted and agree with the reviewer. It was observed that the production of ethanol from xylose is very low when glucose was present. In the cases that xylose was high and glucose concentration was low, most of ethanol was produced from glucose, resulting in low titres. This relation is much more important when considering the production of xylitol. The focus of this work was the industrial substrates and the ability of the cells to ferment them, and not exactly the mechanisms of co-assimilation. However, we presented a model of carbon distribution in the main metabolic route when the cells are co-assimilating glucose and xylose, showing the dependence of redox power supply in the form of NAD(P)H from glucose to support the first reaction of xylose assimilation, showed in Figure 2. However, high glucose resulted in catabolic repression in the way that xylose is untouched when glucose still present.

3 The potential and costs of this strain for industrial application should be dissussed.

RESPONSE: We agree with the reviewer that economic analysis is essential to determine the feasibility of using this yeast for ethanol and xylitol production. However, it was not the objective of this work to carry out an economic analysis because we focused on the possibility of using a mixture of different substrates for this purpose. We intend to consolidate the data for future economic and environmental analysis studies with the proposed system.

4  This manuscript point out " a promising yeast for ethanol and xylitol production". But what the logical linkage between ethanol and xylitol? Is the strain possess outstanding ability for xylose utilization? This manuscript did not show some common features of the strain benified for ethanol and xylitol production.

RESPONSE: In this work, we tried to connect the results with the idea of a biorefinery concept, taking advantage of the residues generated from the production of crystal sugar and fuel ethanol, mainly molasses and cane bagasse. In this sense, using new yeasts can align new process arrangements, such as the co-fermentation of hydrolysates and first-generation substrates. Most of the literature about molasses utilization for ethanol production uses S. cerevisiae, which is the best ethanol producer but not harbors additional metabolic capacities like the production of xylitol and even the production of ethanol from xylose. Thus, by using alternative yeast like M. caribbica or S. passalidarum, the production plants can easily utilize different substrates, especially when fermentation off-season when molasses bagasse still available. Hydrolysates of the cellulose biomass as well as pectic biomass could be mixed to molasses to produce ethanol even at low yield in the period when no ethanol is produced whatsoever. In parallel, a flow of xylose-rich hemicellulose could be used for the production of xylitol. Therefore, we are not proposing the use of M. caribbica to replace S. cerevisiae during the sugarcane season. Instead, we report an alternative for the production of ethanol and xylitol. Anyway, it seems far from the industrial scale and we are completely aware that further studies are necessary to fulfil the minimal requirements for industrial applications.

Reviewer 2 Report

the introduction was improved with some information regarding to M. carribica 87-100. in my opinion this inodrmation are necessary to complete the introduction.

There are still mistakes regarding the form: example line 238: S. cerevisiae not Saccharomyces cerevisiae. Check in all document the correct form of species.

Author Response

The introduction was improved with some information regarding to M. carribica 87-100. in my opinion this inodrmation are necessary to complete the introduction.

RESPONSE: thank you.

There are still mistakes regarding the form: example line 238: S. cerevisiae not Saccharomyces cerevisiae. Check in all document the correct form of species

RESPONSE: checked